# Emerging Applications of *Chlorella* sp. and *Spirulina* (*Arthrospira*) sp.

**DOI:** 10.3390/bioengineering10080955

**Published:** 2023-08-11

**Authors:** Ana P. Abreu, Rodrigo Martins, João Nunes

**Affiliations:** 1Association BLC3—Technology and Innovation Campus, Centre Bio R&D Unit, 3405-155 Oliveira do Hospital, Portugal; rodrigo.martins@blc3.pt (R.M.); joao.nunes@blc3.pt (J.N.); 2BLC3 Evolution Lda, 3405-155 Oliveira do Hospital, Portugal

**Keywords:** microalgae, cyanobacteria, innovative, food, feed, hypoxia, photosensitizers, fuel cells, microrobots, biopolymers, bioremediation, biofuels, aquiculture, bioeconomy, wound healing, cancer

## Abstract

*Chlorella* sp. and *Spirulina* (*Arthrospira*) sp. account for over 90% of the global microalgal biomass production and represent one of the most promising aquiculture bioeconomy systems. These microorganisms have been widely recognized for their nutritional and therapeutic properties; therefore, a significant growth of their market is expected, especially in the nutraceutical, food, and beverage segments. However, recent advancements in biotechnology and environmental science have led to the emergence of new applications for these microorganisms. This paper aims to explore these innovative applications, while shedding light on their roles in sustainable development, health, and industry. From this state-of-the art review, it was possible to give an in-depth outlook on the environmental sustainability of *Chlorella* sp. and *Spirulina* (*Arthrospira*) sp. For instance, there have been a variety of studies reported on the use of these two microorganisms for wastewater treatment and biofuel production, contributing to climate change mitigation efforts. Moreover, in the health sector, the richness of these microalgae in photosynthetic pigments and bioactive compounds, along with their oxygen-releasing capacity, are being harnessed in the development of new drugs, wound-healing dressings, photosensitizers for photodynamic therapy, tissue engineering, and anticancer treatments. Furthermore, in the industrial sector, *Chlorella* sp. and *Spirulina* (*Arthrospira*) sp. are being used in the production of biopolymers, fuel cells, and photovoltaic technologies. These innovative applications might bring different outlets for microalgae valorization, enhancing their potential, since the microalgae sector presents issues such as the high production costs. Thus, further research is highly needed to fully explore their benefits and potential applications in various sectors.

## 1. Introduction

Microalgae are a large and polyphyletic group of O_2_-evolving photosynthetic microorganisms, mostly aquatic, comprising prokaryotic cyanobacteria and eukaryotic members [1]. Estimates of the world microalgal production are around 50.000 t/year [2], *Chlorella* sp. and *Spirulina* sp. accounting for more than 90% of the total microalgal biomass production [3].

The *Chlorella* sp. market is expected to grow at a Compound Annual Growth Rate (CAGR) of 6.3% from 2021 to 2028, reaching USD 412.3 million by 2028 [4]. In terms of value, the Spirulina market is much larger and is expected to reach USD 1.1 billion by 2030, at a CAGR of 9.4% from 2023 to 2030 [5]. For both markets, the main growth drivers include: (1) the consumers’ greater tendency toward a protein-rich diet; (2) increasing awareness for health and wellness; (3) the growth of the nutraceutical industry; (4) an increase in vegetarianism; (5) a growing demand for natural food colors and other microalgal sourced products, such as omega-3 fatty acids; and (6) the development of innovative Chlorella food and beverage products, and products that include Spirulina as an ingredient.

Europe held the largest share of the overall Chlorella market in 2021 [4], where it is cultivated in closed production systems (fermenters or photobioreactors) [6,7,8,9,10,11]. Light can support the growth of *Chlorella* sp. in the photobioreactors (autotrophic conditions), while sugar-based growth is done in the dark inside fermenters (heterotrophic nutrition). The heterotrophic mode for the cultivation of *Chlorella* sp. is expected to grow significantly in the next few years due to higher productivity and a lower risk of contamination, water consumption, and use of space [4]. As photosynthetic organisms, microalgae have the ability to convert light energy into chemical energy. However, many species, such as *Chlorella* sp., *Spirulina* (*Arthrospira*) sp., can also assimilate and oxidize organic carbon molecules, extracting energy from them [1]. This process can occur both in the absence of light, a condition known as heterotrophy, or in the presence of light, referred to as mixotrophy and photoheterotrophy. Although North America is expected to hold the largest share of the Spirulina market in 2030, the market in Europe is expected to register the fastest growth until 2030 [5]. Spirulina is cultivated in raceways, in autotrophic conditions, normally inside greenhouses in Europe and outdoor pounds elsewhere in the world [11,12,13].

*Chlorella vulgaris* held the largest share of the overall market in 2021, but the *Chlorella pyrenoidosa* or *Chlorella sorokiniana* segment is expected to grow significantly in the next few years [4]. Presently, *Chlorella* sp. is a genus from the class Trebouxiophyceae, phylum Chlorophyta, and kingdom Plantae [14]. The taxonomy of *Chlorella* sp. has been evolving for decades and currently represents a group of morphologically similar species of polyphyletic origin rather than a natural genus [15]. A total of 14 species are now assigned to the genus *Chlorella*, including both *C. vulgaris* and *C. sorokiniana*. Conversely, *C. pyrenoidosa* is no longer a valid name and most strains formerly identified as *C. pyrenoidosa* have been renamed, although we can find *C. pyrenoidosa* strains in the literature and listed in culture collections; in this case, *C. pyrenoidosa* refers to strains of uncertain taxonomic status, which have not been examined for reassignment yet [15]. In this review, we will consider the name *C. pyrenoidosa* as it appears in the literature or web sources.

Regarding *Spirulina* sp., a clarification is necessary. Both *Spirulina* sp. and *Arthrospira* sp. include cyanobacterial species very similar to each other, but the two genera are taxonomically distinct [16]. Many species listed in the past as *Spirulina* sp. have more recently been included in the genera *Arthrospira* sp., comprising all those grown commercially and sold as Spirulina [16,17]. Therefore, the trade name continues as Spirulina with no italics. The most important species of *Arthrospira* sp. exploited for commercial mass cultivation include *Arthrospira maxima*, *Arthrospira fusiformis*, and *Arthrospira platensis*.

In this review, species names are referred to as in the original work, irrespective of the taxonomic changes that have occurred, or as “*Spirulina* (*Arthrospira*) sp.”. With reference to commercial names, they are designated by Chlorella and Spirulina.

This review aims to provide a comprehensive understanding of the current and potential applications of *Chlorella* sp. and *Spirulina* (*Arthrospira*) sp. Much of the information is derived from the company websites referenced in market reports [4] and [5], which are also listed in Appendix A. This review article is structured in two parts. The first part describes the current commercial applications of these microalgae, including their use in human food, aquaculture, and cosmetics. The second part highlights innovative and emerging applications, such as animal feed, agriculture, pharmaceuticals, biofuels, biopolymers, and bioremediation. To our knowledge, this is the first article of its kind.

## 2. Current Commercial Applications

### 2.1. Human Food and Nutrition

In 2021, the nutraceutical sector dominated the Chlorella market, due to the distinct properties and benefits of Chlorella as a “healthy food” that contributes to a healthy immune system and body [4]. The increasing consciousness regarding health and well-being, as well as the expansion of the nutraceutical industry, highly contribute to maintaining the nutraceutical segment atop the Chlorella global market. Regarding Spirulina, the nutraceutical segment accounted for the largest share of the market, but the food and beverages segment is expected to grow significantly up to 2030, mainly due to the increasing demand for phycocyanin [5].

The current dominance of the nutraceutical sector in both Chlorella and Spirulina markets is justified by their high nutritional value. Although their microalgal composition varies with culture age and cultivation conditions [18,19,20], *Chlorella* sp. and *Spirulina* (*Arthrospira*) sp. are characterized by their high protein content, low fat, suitable amino-acid profile, high concentration of vitamins (including B12), omega-3 and omega-6 fatty acids, minerals (potassium, calcium, magnesium, selenium, zinc, and others), and bioactive compounds (Table 1).

Currently, Chlorella and Spirulina are mainly sold as powder, as well as tablets, extracts, capsules, or flakes [4,5]. In these forms of commercialization, both Chlorella and Spirulina are sold without any further processing of the biomass, which is only collected and dried.

Although both Chlorella and Spirulina are traded as highly valuated dietary supplements, they have been incorporated in ice creams, snacks, muffins, crackers, bars, cookies, pastry cream, bread, smoothies and other beverages, pasta and noodles, yoghurts, jelly gums, and others [25,29,30]. Additionally, it is noteworthy that some companies have actively engaged in the development of food supplements and nutraceutical ingredients with very high added value. For example, AlgoSource has successfully registered Spirulysat^®^ and Spirugrass^®^ [31,32]. The first is an extract renowned for its elevated phycocyanin content, supplemented with polysaccharides, amino acids, and more. On the other hand, Spirugrass^®^ is a biorefining byproduct of Spirulina characterized by its abundance of amino acids, iron, vitamin K, and beta-carotenes.

Chlorophyll is the predominant pigment in *Chlorella* sp., accounting for 1–2% of its dry weight (Table 2). Thus, the incorporation of *Chlorella* sp. into certain processed foods, providing them with functional and nutritional value, imparts an intense green color that may not be suitable for the specific food product. Consequently, some companies have been actively involved in the development of honey/gold and white Chlorella as alternative solutions to meet and enhance the emerging consumption demands [33,34,35,36]. On the contrary, Spirulina contains phycocyanin, a highly valued pigment known for its natural blue coloration in food (Table 2).

Phycocyanin is a blue pigment synthesized by cyanobacteria, including *Arthrospira* sp. and *Spirulina* sp. In the food industry, it is mainly used as a natural coloring agent in confectionery products, frostings, ice cream and frozen desserts, gelatin, dessert coatings and toppings, beverage mixes and powders, custards, yogurts, puddings, cottage cheese, breadcrumbs, and ready-to-eat cereals [5]. Additionally, it displays antioxidant activity. The phycocyanin market is projected to reach USD 279.6 million by 2030, growing at a CAGR of 28.1% from 2023 to 2030 [37]. This market growth is driven by the increasing adoption of phycocyanin for nutraceuticals and the rising demand for natural blue colorants. Phycocyanin is one of the few natural blue colorants approved in the U.S., Europe, and Asia.

**Table 2 bioengineering-10-00955-t002:** Pigment content of *Chlorella* sp. and *Spirulina* (*Arthrospira*) sp. (mg g^−1^ dw) [25,30,38,39,40,41,42,43,44].

Pigment	*Chlorella* sp.	*Spirulina* sp.
**Chlorophylls**	1.16–24.0	3.01–17.0
Chlorophyll-a	0.25–18.3	2.7–10.8
Chlorophyll-b	0.07–6.81	0.21–0.42
**Pheophytins**	9.73–26.93	8.24–13.49
Pheophytin-a	2.31–5.64	nr
**Carotenoids**	0.24–8.21	0.23–6.5
** Carotenes**	nr	nr
β-carotene	0.007–7.18	0.02–2.5
** Xanthophylls**	nr	2.5–4.7
Astaxanthin	0.25–6.8	0.095–0.72
Canthaxanthin	0.67–1.17	0.44–0.65
Lutein	0.052–13.8	0.12–1.03
Violaxanthin	0.010–0.037	nr
Zeaxanthin	0.074–7.00	0.028–2.0
**Phycobiliproteins**		
Phycocyanin	Absent	95–251

nr—not referenced.

Carotenoids are lipophilic compounds found in *Chlorella* sp. and *Spirulina* sp., which have seen an increased growth in market potential through their uses in foods, pharmaceuticals, cosmetics, animal feed, and as dietary supplements [45]. For instance, carotenoids (β-Carotene) derived from *Chlorella* sp. are sold at USD 300–700 per kg in the open market [46]. Carotenoids such as lutein have also shown great health-promoting properties such as anti-inflammatory properties, and it has been shown that this bioactive pigment can promote or prevent age-related macular disease which is the leading cause of blindness and vision impairment [47].

### 2.2. Aquaculture and Aquarists

In 2020, the total aquaculture animal production reached 87.5 million tonnes, worth USD 264.8 billion, and it is expected to reach 106 million tonnes in 2030 [48]. Sustainable and innovative aquaculture development remains critical to supply the growing demand for aquatic animal foods. To support this trend, the exploitation of alternative, non-traditional protein and oil sources, such as microalgae, for aquafeeds and feeding is necessary. In fact, the nutritional composition and profile of both *Chlorella* sp. and *Spirulina* (*Arthrospira*) sp., as outlined in the preceding Section 2.1, offer advantages when used as a supplement in fish and other aquatic animals.

In aquaculture, microalgae can be introduced in two distinct stages: as food for zooplankton, which then serves as nourishment for fish and their larvae; and incorporated into feed for the nutrition of adult fish and as a substitute for fishmeal or fish oil. When it comes to feeding zooplankton, aquaculture companies typically have their own microalgae production systems. Regarding the incorporation of microalgae in fish feed, feed companies are the main driving force behind change. However, there is not much information available on this topic. The potential of introducing *Chlorella* sp. and *Spirulina* (*Arthrospira*) sp. into fish feed is recognized due to their known benefits. In a similar area, some companies are already producing food for aquarium fish with Chlorella or Spirulina [49,50].

Microalgae species such as *Chlorella* sp. are typically used in finfish hatcheries as a food source for rotifers or *Artemia* sp. [51]. This method results in the bioaccumulation of essential nutrients such as fatty acids, amino acids, carotenoids, vitamins, and minerals from the microalgae, thereby boosting the nutritional value of zooplankton. Zooplankton that has been fortified with microalgae has shown potential as an effective enrichment strategy for promoting the growth and survival of fish and shellfish larvae. For instance, the continuous feeding of loach larvae (*Misgurnus anguillicaudatus*) with live *Moina* sp. and *Daphnia* sp., fortified with *Chlorella* sp., has led to improved growth and survival rates. Similarly, African sharptooth catfish (*Clarius gariepinus*) exhibited enhanced growth and survival rates when fed a combination of microalgae and zooplankton (including *Chlorella* sp. and *Moina micrura*). The survival and growth performance of freshwater carp (*Catla catla*) significantly improved when fed *Cyclops* sp. enriched with *A. platensis*. Fish that were fed *M. micrura* fortified with *Chlorella* sp. showed a significant improvement in specific growth rate and survival. Furthermore, the highest growth and survival rates of *Betta splendens* were observed in fish that were fed copepod fortified with *Chlorella* sp.

*Chlorella* sp. and *Spirulina* (*Arthrospira*) sp. have the potential to replace fish oil and fish meal in diets, enhancing growth and meat quality, stimulating immunity, and improving the pigmentation in various fish species [26,51,52,53,54,55]. They promote growth in a variety of species, including carp, tilapia, Asian seabass, and Nile tilapia. In addition to growth enhancement, these microalgae have been proven to stimulate the immune systems of various fish species [26,51,52,53]. More precisely, *Chlorella* sp. and *Spirulina* (*Arthrospira*) sp. stimulate phagocytosis, increase white blood cell count, and enhance the expression of cytokine genes, playing a crucial role in immune response. *Spirulina* (*Arthrospira*) sp., rich in carotenoids, enhances pigmentation in fish and neutralizes the toxicity of heavy metals, particularly copper and arsenic [51,52,55]. It also improves reproductive performance by increasing egg production and hatching rates [52]. Moreover, in the realm of aquaculture, there exists a widely accepted understanding that the survival rate of fish larvae experiences a remarkable surge when nurtured using live feeds rich in carotenoids, such as rotifers, *Artemia* sp., and copepods [56].

### 2.3. Cosmetics and Skin Care

Bioactive compounds and lipids derived from microalgae are increasingly being recognized as potential alternatives to conventional synthetic ingredients in the cosmetic and skin care sectors [57,58]. As such, extracts from microalgae or specific bioactive compounds can be integrated into the production of a wide range of cosmetic products, including eyeliners, lipsticks, eye shadows, moisturizers, facial cleansers, shampoos, sunscreens, and beauty masks.

The bioactive compounds most commonly utilized in cosmetics include carotenoids, polysaccharides, peptides, and vitamins (Table 3). Bioactive compounds from *Chlorella* sp. and *Spirulina* (*Arthrospira*) sp., with their antioxidant and free radical scavenging abilities, are valuable in cosmetics and skincare products for their anti-aging and wrinkle-reducing potential [59,60]. Carotenoids and peptides offer excellent UV protection in creams and sunscreens, while polysaccharides are ideal for moisturizing purposes, helping to maintain the skin’s water barrier and oil balance.

Microalgal lipids play a crucial role in cosmetics, serving various functions [58,61]. They are used as moisturizing agents, emollients, surfactants, emulsifiers, texturizers, color and fragrance carriers, preservative carriers, and bioactive ingredients. The types of microalgal lipids commonly used in cosmetics include triacylglycerides, waxes, ceramides, phospholipids, sterols, as well as hydrogenated, esterified, and oxidized lipids. Each of these lipids bring unique properties to cosmetic formulations, making them versatile and valuable ingredients in the industry.

With the increasing demand for safe and eco-friendly cosmetics and skin care products, ingredients derived from *Chlorella* sp. and *Spirulina* (*Arthrospira*) sp. are set to take on a significant role in the industry. Their potential to provide efficient and sustainable alternatives makes them an appealing option for both cosmetic manufacturers and consumers in search of innovative and conscientious solutions. In this regard, major companies in the cosmetic industry have been incorporating *Chlorella* sp. and *Spirulina* (*Arthrospira*) sp. into their products, especially creams and serums (Table 4).

Furthermore, it is worth noting that certain companies have been actively involved in the creation of exclusive or even patented ingredients for use in cosmetics. For instance, Algenist created the Blue Algae Vitamin C™, an active L-Ascorbic acid with a blue color, from Spirulina [72]. AlgoSource formulated Spiruderm^®^, a liquid Spirulina extract highly concentrated in phycocyanin, used as an active ingredient for skin moisturizing and re-densifying, and fine lines smoothing [73].

## 3. Emerging and Innovative Applications

### 3.1. Animal Feed

After thorough research on the internet, it was found that only a small number of animal feed products currently incorporate *Chlorella* sp. or *Spirulina* (*Arthrospira*) sp. (as exemplified in [74,75]). As a result, this microalgal application has been categorized under the “Emerging and innovative applications” section. Despite the significant potential in introducing Chlorella or Spirulina into animal feed formulations or additives, it is evident that this area is still under active research.

Based on a limited number of studies, it has been observed that incorporating Spirulina into poultry and swine diets does not have a significant impact on animal productivity and product quality, while also holding the potential to replace less sustainable protein sources such as fishmeal [76]. Furthermore, Spirulina has demonstrated numerous benefits for poultry, including a decreased cholesterol, an improved growth performance, and an enhanced immune system, among others [77]. It also increases high-density lipoproteins in chickens, as well as oxidant capacity, oxidative stability, and glutathione peroxidase. Similarly, incorporating *C. vulgaris* in the diet of New Zealand white rabbits improved their growth performance parameters, nutrient utilization, intestinal efficacy, and antioxidants [78]. Further research is essential to fully explore the benefits and potential implications of incorporating Chlorella and Spirulina in animal feed. This will help optimize their use for animal health, productivity, and sustainability in the industry.

### 3.2. Agriculture

Nowadays, worldwide agriculture practices are extremely dependent on synthetic fertilizers and pesticides, with environmental and health consequences. Microalgae can contribute to a change in this scenario, by fixing nitrogen in the soil, cycling phosphorus and other nutrients, promoting plant growth, ameliorating rooting, stabilizing the soil, and providing more tolerance to drought and salinity, enhancing pathogen resistance and other critical agricultural needs [79,80,81].

As in Section 3.1, the potential application of *Chlorella* sp. and *Spirulina* (*Arthrospira*) sp. in agriculture was categorized under the “Emerging and innovative applications” section, due to the lack of products found on internet. However, these microalgae species have shown promising potential in various agricultural applications as biofertilizers and biostimulants.

The application of *C. vulgaris* as a biofertilizer in tomato (*Solanum lycopersicum* L.) cultivation resulted in enhanced tomato plant growth and yield [82]. This study compared the application of *C. vulgaris* by a foliar spray method and a soil drench method with and without cow dung and control (untreated). A mixed treatment of cow dung with *C. vulgaris* showed enhanced growth compared to the other groups: a fruit length and diameter of about 11.7 cm and 16.7 cm, respectively; an amount of seeds/fruit of about 153 g; and a seed weight/fruit of about 6.5 g. The nutritional composition and shelf-life of fruits were also improved. A foliar spray with C. vulgaris promoted a 26.9% increase in the total length and height of common beans (*Phaseolus vulgaris*) compared to the control (untreated), as well as an increase in dry weight (37.28%) and both protein and carbohydrate contents during the vegetative stage; during the fruiting stage, compared to the control, there was a measured increase in the number of pods per plant (from 3.4 to 5.2), the number of seeds/pod (from 2.6 to 3.5), and the dry weight of the pod (from 0.67 to 0.95 g plant^−1^) [83]. Similarly, a foliar spray application of *C. vulgaris* extract positively influenced the growth of lettuce seedlings, by increasing the fresh and dry weights, chlorophylls, carotenoids, protein content, and ashes at the shoot level [84]. Additionally, the primary and secondary metabolisms of shoots, in particular the nitrogen metabolism, were positively influenced. At the root level, the extract increased dry matter, proteins, and ash content. *C. sorokiniana* stimulated the maize root system in hydroponic conditions, compared to the untreated control, by increasing root area and root volume, total root length and the number of secondary roots; moreover, *C. sorokiniana* improved the development of the root system in nitrogen deficiency conditions [85]. The application of live *Chlorella* sp. significantly increased the *Medicago truncatula* height (11%), leaf size, fresh weight (36%), number of flowers (36%), and carotenoid content (31%) under controlled greenhouse conditions compared to the control [86]. Biofertilizer extracts of *C. vulgaris* and *S. platensis* improved the growth of green gram (*Vigna radiata*), including its shoot and root length, and its weight at the flowering stage [87]. These treatments also improved the plant’s physical characteristics, such as water and oil absorption, and positively affected the soil’s pH, EC, and mineral content. Furthermore, the leaf chemical composition, including N, P, K, total carbohydrates, indole, and phenol, improved in plants treated with these microalgae. The nutritional and mineral composition of green gram seed flour also increased with these treatments.

Recently, *Spirulina* (*Arthrospira*) sp. attracted attention as a sustainable source for agricultural biostimulants [88]. Its potential has been demonstrated on various crops, enhancing their growth traits and stress tolerance. This effect is likely due to a synergy of bioactive molecules such as phytohormones, polysaccharides, vitamins, and amino acids. Therefore, Spirulina-based biostimulants could be a sustainable alternative to chemical fertilizers, pesticides, and growth modulators. For instance, extracts of *S. platensis*, rich in fatty acids, exhibited potential as biostimulants on the initial growth phase of wheat (germination tests) and the yield of wheat and rapeseed (field tests) [89]. Moreover, the Spirulina-based biostimulants were effective against six of the nine fungal pathogen strains tested and reduced the development of pathogens. Evaluating the effect of two biostimulants, *S. platensis* and Egyptian clover (*Trifolium alexandrinum*), on the soil properties, growth, and yield of a pea (*Pisum sativum* L.) plant, it was found that the plant growth and yield were significantly increased when these biofertilizers were applied either individually or in combination [90]. Compared to the control, the highest concentration of combined treatment significantly increased the shoot length, number of leaves, leaf area and number of branches by 145, 200, 300 and 100%, respectively. Furthermore, the combined treatment showed the highest values in the number of pods (8), pod length (12 cm), no. of seeds (13) and dry weight of seeds (21.5 g) compared to control. In addition, the physicochemical properties and mineral status of soil were improved after the application of the biostimulants.

In addition to the benefits presented, microalgae can treat wastewaters, and the biomass that they produce during this process can be utilized as a biofertilizer. For instance, *Chlorella* sp. cells removed the amount ammonium, phosphate, and nitrate from a landfill leachate by 98.7, 92.7, and 56.9%, respectively, and the biomass obtained was used as an agricultural fertilizer [91]. The addition of microalgae biomass to the soil caused positive and significant changes in soil organic carbon, available phosphorus and nitrogen, microbial basal respiration, and microbial biomass carbon. Similarly, an undiluted poultry effluent served as the culture medium for the growth of microalgae, including *C. vulgaris* and *C. protothecoides*, in a semi-continuous (28 days) mode [92]. The remediation rates were 100% for total nitrogen and phosphorus, and over 92% for chemical oxygen demand (COD). The produced biomass was tested as a biostimulant and showed a 147% increase in wheat germination index.

### 3.3. Pharmaceutical Drugs

Microalgae produce a wide range of bioactive compounds that exhibit diverse biological properties, including antioxidant, anticoagulant, anti-inflammatory, antimicrobial, anticancer, and neuroprotective effects among others. These bioactive compounds include proteins, lipids, carbohydrates, vitamins, and a variety of secondary metabolites (Table 5). The potential applications of these bioactive compounds derived from microalgae have attracted considerable attention, given their prospective uses across a variety of sectors such as food, animal feed, cosmetics, and pharmaceuticals [57,93,94,95].

The bioactive compounds from *Chlorella* sp. and *Spirulina* (*Arthrospira*) sp. have the potential to contribute to the development of new drugs for humans and animals. Their diverse biological properties could be harnessed for the creation of innovative treatments for various diseases, including different types of cancers, heart diseases, and viral infections. This highlights the importance of further research into these microalgae and their bioactive compounds as they hold great promise for the future of pharmaceutical science.

### 3.4. Wound-Healing Dressings

Chronic wound healing presents a substantial challenge for both patients and healthcare providers. This issue, which primarily involves conditions such as ulcers, diabetic foot wounds, and burns, necessitates suitable treatment to prevent complications such as scarring, discoloration, or joint contractures. With the global geriatric population on the rise, this significant health and social issue is projected to become even more prevalent.

Depending on the type of wound, it is crucial to use appropriate dressing materials and structures that can maintain a moist environment at the wound site while simultaneously protecting it from contaminants. In recent years, both academic and industrial sectors have been making concerted efforts to develop a variety of wound dressings suitable for wound healing, specifically exploring the use of different microorganisms, such as microalgae, due to their beneficial bioactive compounds [59,60,103,105,106]. As mentioned in Section 3.3, microalgae contain an extensive variety of bioactive compounds that can promote skin regeneration. As they are hypoallergenic and can be safely applied directly to the skin, wound dressings based on microalgal extracts are attracting significant interest. Innovative solutions have been proposed, based on *Chlorella* sp. and *Spirulina* (*Arthrospira*) sp. and some are summarized in Table 6.

### 3.5. Overcoming Hypoxia in Tissue Engineering and Cancer Therapies

#### 3.5.1. Tissue Engineering

Tissue engineering is a field that aims to repair and regenerate damaged tissues or organs by creating engineered tissue constructs. However, one of the main challenges in this field is ensuring an adequate oxygenation of these constructs, both *in vitro* and post-implantation *in vivo*. Hypoxia can lead to cell death and can compromise the effectiveness of the constructs. Moreover, it can aggravate the conditions of many oxygen-deficiency-aggravated diseases, such as cancer, ischemic heart disease, and chronic wounds. To address this, researchers are exploring the development of “breathing” biomaterials that could provide adequate oxygen and vascularization of immobilized cells. Microalgae could play a crucial role in addressing hypoxia in tissue engineering and regenerative therapies [105].

Microalgae could provide a sustained supply of O_2_ to the engineered tissues, improving local oxygenation, and preventing hypoxia-induced cell death and dysfunction [118,119]. In addition, microalgae extracts have antitumor, anti-inflammatory, antibacterial, and antioxidant effects. In this context, numerous research efforts have been undertaken involving the incorporation of *Chlorella* sp. and *Spirulina* (*Arthrospira*) sp. into bioengineered tissues.

A “breathing” biohybrid material was developed by incorporating *C. sorokiniana* into an engineered bioartificial pancreas [120]. The O_2_ generated by *C. sorokiniana* compensated for the O_2_ requirements of encapsulated mouse pancreatic islets and provided proper insulin secretion. A respiratory support system, leveraging the ability of *C. vulgaris* to provide a constant supply of O_2_, was developed to extend the viability of rat pancreas cells after cardiac death for transplantation purposes [121]. The transplantation of these rat cells, which were preserved under photosynthetic respiratory support, resulted in the survival of all treated rats—an outcome that was unattainable when using pancreases stored in static cold conditions. An efficient transdermal delivery method based on microneedles encapsulating *C. vulgaris* (CvMNs) was developed for the O_2_ supply in diabetic wound healing [122]. The CvMNs have a polyvinyl acetate (PVA) substrate and gelatin methacryloyl (GelMA) tips. Upon application to the diabetic wound, the PVA base dissolves quickly, while the GelMA tips remain in the skin. Both *in vitro* and *in vivo* results showed that, under near-infrared light, *C. vulgaris* produces O_2_, promoting fibroblast proliferation and angiogenesis. Concomitantly, the antioxidant compounds from the microalga inhibit the inflammation, accelerating the wound healing. In addition, *C. vulgaris* could remain alive for at least 6 days, which would ensure the continuous production of O_2_. A similar work using microneedles and encapsulating *C. vulgaris* (CvMNs) was elaborated by Wang et al. [123] to supply O_2_ for antiphotoaging treatment. Dissolved O_2_ can reverse photoaged skin, but the availability of equipment (e.g., high-pressure O_2_) and poor gas diffusion into the skin have limited its therapeutic efficacy. The CvMN patch could deliver living *C. vulgaris* to the deeper layers of the skin for efficient oxygenation, reduction of inflammation, and collagen regeneration, thus effectively reversing photoaging and reducing wrinkles. A living hydrogel with functionalized *C. pyrenoidosa* and *Bacillus subtilis* encapsulation was created for promoting chronic wound healing [124]. The microalga continuously released O_2_ to relieve hypoxia and enhance the survival of *B. subtilis*, while this bacterium eliminated the colonized pathogenic bacteria through releasing antimicrobial agents. Other hydrogels with living *Chlorella* sp. demonstrated the potential *in vitro* and *in vivo* for diabetic wound healing, by producing O_2_, consuming glucose, and depleting reactive oxygen species with the inherent antioxidants [113].

Another strategy that has been developed to provide O_2_ to the tissues is 3D bioprinting, incorporating microalgae to create “living” scaffolds that can produce O_2_. These scaffolds can adapt to irregular-shaped wounds and promote their healing [125]. A living scaffold incorporating *C. pyrenoidosa* was directly printed into diabetic wounds, accelerating significantly the chronic wound closure by alleviating local hypoxia, increasing angiogenesis, and promoting extracellular matrix synthesis [126].

#### 3.5.2. Anti-Cancer Therapy

Photodynamic therapy (PDT) is a promising treatment method that utilizes the synergistic action of a photosensitizing agent, light of a specific wavelength, and oxygen. This combination produces reactive oxygen species (ROS), which can cause oxidative damage to tumor cells. PDT is particularly promising due to its minimal invasiveness and its ability to target treatment to specific areas, reducing the side effects. However, PDT itself can exacerbate hypoxia, leading to a reduced therapeutic effect, resistance, and irreversible tumor metastasis. Thus, O_2_ must be continuously provided directly or indirectly to the tissues. One potential strategy to supply O_2_ is based on microalgal photosynthesis, which can produce oxygen in situ, significantly improving the tumor’s anoxic environment [119,127,128].

Zhou et al. [129] designed a light-triggered oxygen-affording engine to generate O_2_ in situ with high controllability. When a chlorophyll derivate (Ce6) was irradiated by a 635 nm laser, the energy was transferred to the O_2_, produced in large amounts from *C. pyrenoidosa*, generating more singlet oxygen. The created engine, which was biocompatible and degradable, could be switched on and off using the same irradiation as PDT and achieve nearly three times more O_2_ than inorganic oxygen production materials. The study demonstrated that the O_2_ produced by *C. pyrenoidosa* could alleviate hypoxia during PDT therapy and effectively eliminate tumor tissues. Other study demonstrated the dual role of *Chlorella* sp. in tumor killing [130]. The microalga was used as an O_2_ producer to reverse hypoxia and release adjuvants to reverse the immunosuppressive microenvironment, enhancing the abscopal effect afterwards. The treatment with *Chlorella* sp. produced a stronger antitumor immune memory effect and elicited antitumor effects toward established primary tumors (inhibition rate: 90%) and abscopal tumors (75%), controlled the challenged tumors (100%), and alleviated metastatic tumors (90%). Wang et al. [131] reported a way that O_2_ was supplied for PDT: O_2_ was photosynthetically produced by a *Chlorella* sp. strain (reaching 500 µM in a short time) and the excess was collected by perfluorocarbon (PFC) and used to the greatest extent, enhancing the therapeutic effect of PDT. Compared with the system without PFC, the sustainable Chlorella-PFC system could produce up to ten times the therapeutic effect both *in vitro* and *in vivo*. After light stops, *Chlorella* sp. further acted as an adjuvant to promote dendritic cell activation, promoting an antitumor immune response.

### 3.6. Photosensitizers

A photosensitizer (PS) is a molecule that can undergo electronic excitation, i.e., once a particular wavelength of light is absorbed by the photosensitizer, its electrons are temporary excited to higher and more unstable energy levels and will return to intermediate states or to ground state by releasing the excess energy as an electron flow [132]. Thus, microalgal photosynthetic pigments are photosensitizers.

Chlorophylls are the most abundant pigments found in *Chlorella* sp., reaching 1–2% dry weight (Table 2). A similar content of chlorophyll can be found in *Spirulina* (*Arthrospira*) sp., but phycocyanin is the most abundant pigment in this microalga. Chlorophylls are the main responsible compounds for the light energy capture and further conversion into chemical energy [133]. Carotenoids and phycobiliproteins (in cyanobacteria) are accessory light-harvesting pigments that capture and transfer light for the chlorophylls in the reaction centers. Carotenoids also have a photo-protective function as they protect chlorophyll molecules during strong exposure to radiation and oxygen.

During photosynthesis, the flow of electrons contributes to the synthesis of NADPH and ATP, two crucial molecules for energy storage and transfer [133]. However, the electrical power released in this process can also stimulate molecular oxygen or semiconductors. This can be applied in photodynamic therapy, where the energy is used for medical treatments (Section 3.6.1), or in dye-sensitized solar cells, where it is used to generate electricity (Section 3.6.2).

#### 3.6.1. Antimicrobial and Anticancer Photodynamic Therapy

The PDT is a promising treatment method that utilizes the synergistic action of a photosensitizing agent, light of a specific wavelength, and oxygen. This combination produces ROS, which can cause oxidative damage to targeted cells, including bacteria, fungi, parasites, viruses, and cancer cells, without significantly harming healthy tissues [134,135]. Briefly, the PS is administrated into the targeted cells and photoexcited at a particular wavelength, generating an electron flow that is transferred to oxygen molecules (O_2_), producing singlet oxygen (^1^O_2_) or superoxide (O_2_^−^). PDT is particularly promising due to its minimal invasiveness and its ability to target treatment to specific areas, which reduces the side effects. Moreover, it does not contribute to the development of microbial resistance, making it a potential alternative or supplement to traditional antimicrobial therapies. PDT has demonstrated efficacy against a broad range of infections and diseases, including dental infections, skin disorders, and various types of cancer. This makes it a versatile and promising approach in the field of medical treatment.

Natural PSs, as opposed to synthetic compounds, demonstrate several advantages including lower toxicity to healthy tissues, tissue specificity, a broad spectrum of target pathogens, a lower incidence of side effects, reduced dark toxicity, and the potential to overcome antimicrobial resistance [132,136]. Despite these promising attributes, further research is needed to fully explore and understand the potential of these natural compounds in PDT.

The derivates of chlorophylls and bacteriochlorophylls are among the natural PSs suitable for PDT and include pheophorbide-a (PPBa), pheophorbide-b methyl ester, 13(2)-hydroxyl (13(2)-S) pheophorbide-a methyl ester, 13(2)-hydroxyl (13(2)-R) pheophorbide-b methyl ester, pyro pheophorbide methyl ester, and chlorin e6 [137,138].

Pheophorbide-a (Pha) derived from *S. maxima* was covalently attached to acetylated xylan extracted from chestnut sawdust to form the xyl-Pha conjugate used in the nanoparticles for tumor-targeted PDT [139]. After red light irradiation (630–660 nm at 75 J/cm^2^), the xyl-Pha nanoparticles showed cytotoxic activity against HT-29 human colorectal cancer cells. Similarly, a Tryptamine–Pha conjugate, derived from Spirulina powder, showed significant photocytotoxicity (IC50 = 695 nM) towards lung A549 cells upon irradiation with red light [140]. Nanoparticles of N-vinylpyrrolidone amphiphilic copolymers and Pha processed from *Spirulina* sp. were evaluated as promising PSs; when irradiated with red light (λ = 660 nm), the *in vitro* phototoxic effect of the nanoparticles in HeLa cells exceeded by 1.5–2 times that of the reference dye chlorin e6 trisodium salt—one of the most effective PSs used in clinical practice [141]. A PDT based on chlorin e6 (Ce6) from *Spirulina platensis* not only confirmed the suppression of a mouse melanoma on the mouse left flank, but also the induction of systemic effects in non-irradiated tumors on the right flank, where no Ce6-PDT was given, by triggering an immune response [142]. A derivative of Ce6 from *S. platensis*, the Ce6-trimethylester, was evaluated in the potential treatment of HeLa cells, reducing cell viability after irradiation with a 650 nm laser at 1.2 J/cm^2^/min [143]. Another study evaluated the cytotoxic effects of Ce6 from *S. platensis* on B16F10 melanoma cells [144]. Higher concentrations of Ce6 (up to 192 µM) showed minimal dark cytotoxicity on B16F10 cells (IC50 519.6 µM), but when the cancer cells were exposed to light of 660 nm at 50 mW for 1 min and 40 s, a pronounced light-induced cytotoxicity was observed with a 27-fold lower concentration against the tested cells (IC50 18.9 µM). Furthermore, the researchers confirmed the potential of Ce6-PDT to inhibit the growth of B16F10 melanoma cells *in vivo*, using B16F10 allograft mice. A derivate of Ce6 ((E)-3^2^-(4-methoxyphenyl)-chlorin e6), processed from Spirulina powder, exhibited phototoxicity against HepG2 cells, whose viability was only 10% at 1 μM [145]. The PS was also evaluated on tumor-bearing mice; these tumors were significantly inhibited in the PDT group.

The photodynamic inactivation of *Staphylococcus aureus* and *Candida albicans* was evaluated using eight derivates of chlorophyll as PSs [146]. The PSs containing carboxylic acids and butyl groups were highly effective against both microorganisms, showing significant toxic effects at a concentration of 10 µM and a light dose of 15 Jcm^−2^ and 660 nm. Ce6 derived from *Chlorella ellipsoidea* effectively inhibited the growth of *S. aureus* and *Pseudomonas aeruginosa* after PDT application with 10 μM of Ce6 and a laser of energy density of 20 J/cm^2^ [147]. However, the Ce6-mediated PDT showed a lesser effect on *Escherichia coli* and *Salmonella enterica* serovar Typhimurium even with 20 μM of Ce6 and a laser of energy density greater than 30 J/cm^2^. A Ce6-derivative obtained from *Spirulina* sp. was evaluated as a PS for the treatment of cervical and vaginal low-grade squamous intraepithelial lesions (LSIL), associated with persistent human papillomavirus (HPV) infections [148]. The results showed that the complete remission rates were 88.89% and the HPV remission rates was 94.44% at a 6-month follow-up. There were no recurrence cases, no pain, heaving sensation, scar, or deformity, and no other obvious adverse reactions occurred during the treatment and follow-up.

#### 3.6.2. Dye-Sensitized Solar Cells

Dye-sensitized solar cells (DSSCs) (also known as Bio-sensitized solar cells, BSSCs) are multifaceted structures that convert absorbed solar radiation into electricity. DSSCs are among the 3rd generation of photovoltaic technologies that has attracted much attention in the conversion of solar radiation into electrical power [149]. The sensitizer is the fundamental component of DSSCs, absorbing light and converting it into electricity that will reduce the conductive and transparent oxide film (usually fluorine-doped tin oxide or indium tin oxide) coated on the glass surface [150]. Generally, the photosensitizers can be categorized as follows: natural, metal-complexes, metal-free, and porphyrin-based.

Natural photosensitizers include photosynthetic pigments, proteins, photosystems, and reaction centers [151]. Photosynthetic pigments have been the primary choice for natural sensitizers as they are naturally suited for efficient light harvesting. The reported photoconversion efficiencies vary from 0.001% to 4.6% for sensitizers extracted from microalgae.

Phycocyanin extracted from *S. platensis* was used as a photosensitizer to electrify a silver (Ag)-doped TiO_2_ film with a photovoltaic conversion efficiency of around 1.2% [152]. The short-circuit photocurrent density (J_SC_) of the cell was about 3.0 mA/cm^2^ and the open-circuit photovoltage (V_OC_) was 0.75 V under an illumination intensity of 40 mW/cm^2^. Similarly, a nano-TiO_2_ film sensitized with phycobiliprotein extracted from *Spirulina* sp. exhibited a photovoltage of 273 mV and a current of 4.2 mA [153]. A DSSC using chlorine-e6 (Chl-e6) derived from chlorophyll extracted from *Spirulina* sp. adsorbed on a nanocrystalline TiO2 film electrode was developed [154]. The J_SC_ value obtained was 0.305 mA cm^−2^, the V_OC_ was 426 mV, and the calculated fill factor (FF) was 45.0%. β-carotene molecules overproduced and purified from a *Chlorella* sp. strain were adsorbed on the surface of a nanocrystalline TiO_2_ film electrode and exposed to simulated solar radiation; the solar cells sensitized by β-Car produced a J_SC_ value of 0.185 mA/cm^2^, a V_OC_ value of 0.230 V, maximum power output of 1.8 μW/cm^2^, a calculated FF value of 0.303 and an efficiency of 0.022% [155]. In comparison, solar cells sensitized with a crude extract of photosynthetic pigments were also built up and exhibited a J_SC_ value of 0.012 mA/cm^2^, a V_OC_ value of 0.230 V, maximum power 0.975 μW/cm^2^, a calculated FF value of 0.352, and a very poor efficiency of 0.001%, supporting the role of the β-carotene pigments as photosensitizers.

### 3.7. Microrobots

For the successful implementation of PDT, chemotherapy, and immunotherapy, it is crucial to ensure a continuous supply of O_2_ to the damaged or cancerous tissues, as explained in Section 3.5. Additionally, it is essential to precisely deliver therapeutic drugs, including the photosensitizer (Section 3.6.1), to the specific target area. Therefore, the use of microcarriers becomes essential for this purpose.

Microrobots (or microswimmers) can be defined as active microcarriers made of photosynthetic microorganisms. Microrobots have shown significant potential to conduct microscale tasks such as drug delivery, cell manipulation, microassembly, and biosensing using manual control [156]. Their size and controllability are key features for controlled navigation in hard-to-reach cavities of the human body, making them promising miniaturized robotic tools to diagnose and treat diseases in a minimally invasive manner [157]. Due to their potential, microrobots have received extensive attention in the past few decades, while the in-depth research of micro and nano processing technology has promoted the further development of microrobots [158].

In the scope of microrobots, *Spirulina* (*Arthrospira*) sp. Has gained extensive interest due to its morphological features such as the helical angle, helix diameter, and body length [159]. The spiral morphology and corkscrew motion of *Spirulina* (*Arthrospira*) sp. Enables its use as a microswimmer to transport therapeutics to pulmonary capillaries or in neuronal regenerative therapies. For instance, Yan et al. [157] were able to create helical microswimmers from *S. platensis* through a dip-coating process using iron oxide nanoparticle suspensions. The innate properties of the microalgae species allowed *in vivo* fluorescence imaging and remote diagnostic sensing without the need for any surface modification. Similarly, molecular cargos using cell-based helical magnetic microswimmers were engineered from *S. platensis* for targeted molecular delivery in an intestinal tract mimicry [160]. The loaded molecules could be released from the swimmer through host degradation and/or concentration gradient-driven diffusion and, after the loading-release, the bioactivity of the guest molecules remained intact. A wirelessly controllable *S. platensis*-Fe_3_O_4_-tBaTiO_3_ micromotor was engineered by Liu et al. [161] for precise neural electrical stimulation at the single-cell level, a promising solution for patients with neurotrauma and neurodegenerative diseases. The microalga was coated with magnetic Fe_3_O_4_ and piezoelectric BaTiO_3_ nanoparticles, and the biohybrid micromotors could precisely induce the differentiation of the targeted neural stem-like cell by converting ultrasonic energy to an electrical signal in situ. To treat pathogenic bacterial infection, Xie et al. [162] developed a microswimmer consisting of a magnetized (Fe_3_O_4_ nanoparticles) *Spirulina* sp. (MSp) matrix coated with polydopamine (PDA). Noninvasive and real-time image tracking (by photoacoustic imaging) of the MSp-PDA microswimmer was accomplished and its photothermal therapy for the theranostics of pathogenic multi-drug-resistant *Klebsiella pneumoniae* infection was also demonstrated. In order to design a drug delivery system to fight against metastatic breast cancer, a *S. platensis* (Sp) strain was used as a natural carrier of the chemotherapeutic doxorubicin (DOX) [163]. The Sp-DOX carriers exhibited ultrahigh drug loading efficiency and sustained release, resulting in a significantly enhanced therapeutic efficacy on the lung metastasis of 4T1 breast cancer. Moreover, the rich chlorophyll gives Sp-DOX an excellent fluorescence imaging capability for noninvasive tracking and real-time monitoring *in vivo*. Zhong et al. [164] fabricated a multifunctional nanoswimmer system for tumor-targeted imaging and therapy based on magnetically engineered *S. platensis*. The microalgae generated O_2_ in situ and the released chlorophyll, acting as a photosensitizer, produced cytotoxic ROS upon laser irradiation. Moreover, the chlorophyll-based fluorescence and photoacoustic imaging allowed the monitorization of the tumor therapy and tumor microenvironment. In another study, a living hydrogel containing carboxymethyl chitosan-coated *S. platensis* was prepared to keep the O_2_ generation capability of the gel while promoting its adhesion to the infected wound in a mouse model [165]. Additionally, chlorophyll, acting as a photosensitizer, produced ROS when exposed to a 650-nm laser, leading to the photodynamic eradication of *S. aureus*-infected skin. A similar work was conducted by Hu et al. [166]. They loaded berberine (BBR, a quorum sensing inhibitor, and antibacterial agent) into a natural living *S. platensis* (Sp) to form a bioactive hydrogel (BBR-Sp gel) in combination with carboxymethyl chitosan/sodium alginate. Under laser irradiation, the BBR-Sp gel could constantly release BBR and produce ROS, resulting in a synergistic quorum sensing inhibition against methicillin-resistant *S. aureus* (MRSA)-combined chemo-photodynamic therapy. The BBR-Sp gel also relieved hypoxia, destroyed biofilm formation, and down-regulated the expression of virulence factors. The BBR-Sp gel accelerated MRSA-infected diabetic wound healing in mice by promoting angiogenesis, skin regeneration, and suppressing the inflammatory response.

Similar work has been conducted with *Chlorella* sp. cells. Living *C. vulgaris* cells were engineered for an in situ generation of O_2_ to relieve tumor-bearing mice hypoxia, enhancing radiotherapeutic effect [167]. Furthermore, a cascade phototherapy was fulfilled by the chlorophyll released from *C. vulgaris*’s combined thermal effects under 650 nm laser irradiation. The treatment was validated in an orthotropic breast cancer mouse model, revealing its prominent antitumor and antimetastasis efficacy in hypoxic-tumor management. The engineered *C. vulgaris* exhibited excellent fluorescence and photoacoustic imaging properties, allowing the self-monitoring of the tumor therapy and tumor microenvironment. The surface of a *Chlorella* sp. strain was engineered for the targeted delivery of DOX and to alleviate hypoxic tumor microenvironment for enhanced chemotherapy and immunotherapy [168]. The *Chlorella* sp. cells were coated with a macrophage membrane (M-Chl) for camouflage, to enhance tumor accumulation and the retention of *Chlorella* sp. due to the inflammatory homing effects of a macrophage membrane. A continuous production of O_2_ was achieved for at least 6 days, resulting in an efficient downregulation of tumor HIF-1α. *In vivo* antitumor chemotherapy showed that DOX exhibited unsatisfactory antitumor effects in melanoma mice, but its therapeutic effect significantly improved in the mice treated with M-Chl for tumor oxygenation. In a similar work with *C. pyrenoidosa*, magnetic microrobot multimers were engineered for targeted DOX delivery [169]. The microrobot multimers exhibited diverse swimming modes including rolling and tumbling with high maneuverability, and the rolling dimer’s velocity could reach 107.6 μm/s (~18 body length/s) under a 70 Gs processing magnetic field. Chemotherapy experiments toward HeLa cancer cells confirmed the high DOX loading capacity and low cell toxicity of the microcarriers. In fact, the biodegradability and toxicity of microrobots should always be addressed, to ensure their safe removal from the human body. A comprehensive review article by Li et al. [170] provides valuable insights into this topic.

### 3.8. Biofuels

Biofuels are an essential part of the world’s transition towards sustainable and renewable energy sources. They represent a viable solution for reducing greenhouse gas emissions (GHG), particularly in sectors that are difficult to decarbonize, such as aviation and transportation. Moreover, biofuels can be integrated into existing fuel infrastructure with minimal changes, making them a practical and cost-effective solution. However, sustainable practices in biofuel production are necessary to maximize their benefits and minimize potential negative impacts, such as competition with food and feed crops. A growing focus on ensuring sustainability, meanwhile, has led to increasing attention being paid to advanced liquid biofuels that do not directly compete with food and feed crops and that avoid adverse sustainability impacts [171].

Microalgae have gained significant attention in recent years as a promising feedstock for biofuels. This interest is primarily due to their potential to produce lipids and carbohydrates, which can be converted into a variety of biofuels such as biodiesel, bioethanol, biogas, biomethane, biosyngas, bio-oil, and biohydrogen [172]. Briefly, biodiesel can be derived from the lipids of microalgae through a process known as transesterification, while bioethanol is commonly produced by fermenting the carbohydrates found in microalgae. Biogas and biomethane can be generated from the anaerobic digestion of microalgal biomass. Additionally, the gasification of this biomass can result in bio syngas, and pyrolysis can convert microalgal biomass into bio-oil. As for biohydrogen, it can only be produced by specific microalgae species, one of which is *Chlorella* sp. In addition to its potential for the production of biofuels, microalgae can grow in various environments such as saltwater and wastewater, saving arable land and freshwater for food crops. They grow quickly and efficiently, yielding more biofuel per area than land plants, and help reduce greenhouse gases by absorbing CO_2_ from the air or industrial emissions.

Given the extensive body of work published on the production of biofuels from microalgae, particularly from *Chlorella* sp. and *Spirulina* (*Arthrospira*) sp., only a few examples of recent review articles addressing this topic are mentioned here [172,173,174,175,176]. It is also worth noting that these biofuels can be produced in conjunction with bioremediation processes. That is, wastewaters and CO_2_ can be incorporated into the cultivation of *Chlorella* sp. and *Spirulina* (*Arthrospira*) sp. for biofuel production. For example, *C. vulgaris*, when grown in swine wastewater (SW) in outdoor open raceway ponds of 5 m^3^ and supplemented with 3% CO_2_, yielded a total of 21.3% fatty acids, which were trans-esterified for biodiesel production, at a conversion rate of 93.3% [177]. Additionally, the process demonstrated a nutrient removal efficiency from SW of 82.1% for total nitrogen and 28.4% for total phosphorus. Similarly, *C. sorokiniana* was cultivated in a 3000 L pond over a period of 5–7 days using non-sterile dairy effluent, achieving a biomass yield of 1.6 g/L and a lipid content of 25% (*w*/*w*) [178]. The conversion efficiency of the decolorized microalgal biodiesel obtained after transesterification was 40% (*w*/*w*) and the fuel properties were within the required biodiesel specifications.

### 3.9. Microalgae-Assisted Microbial Fuel Cells

Microalgae-assisted microbial fuel cells (MA-MFCs) (also known as algae-photosynthetic microbial fuel cells) can be described as a bio-electrochemical system that generates electric power by exploiting naturally occurring microalgal metabolic processes [179]. In brief, bacteria or microalgae (in suspension or attached as a biofilm) oxide organic matter in the anodic chamber, generating electrons and protons. The electrons flow through an external circuit, resulting in an electrical current that can be harnessed and used, while the protons pass through a proton exchange membrane to the cathode compartment. Here, the protons combine with the electrons and oxygen generated from microalgal photosynthesis to form water.

As a reference, an AA battery-sized MA-MFC constructed from readily available, durable, affordable, and mostly recyclable materials successfully powered the continuous operation of an Arm Cortex M0+ processor for more than six months in a domestic environment, without the need for additional energy storage, artificial lighting, or organic feeding [180]. In a controlled laboratory environment, the prototyped MA-MFC showed a peak power density of 42 mW m^−2^. Similar values have been obtained in MA-MFCs powered by *Chlorella* sp., *Arthrospira* sp., and *Spirulina* sp. (Table 7).

MA-MFCs could be integrated into a variety of promising applications, including wastewater treatment, CO_2_ mitigation, and desalination [179]. A fuel cell device was powered by *Chlorella* sp., used as substrate in the anode and as a live culture in the cathode, and fed by flue gas (10% CO_2_ *v*/*v*) and domestic wastewater [181]. The highest power output generated was 54.48 mW m^−2^, with a maximum current of 3.1 mA, and a stable voltage of 0.852 V. Additionally, biomass productivity in the biocathode was 106.6 mg l^−1^ d^−1^, CO_2_ sequestration was 191 mg l^−1^ d^−1^, and the efficiency of COD removal from the waste stream was 75%. During a 120-day period, the performance of a MA-MFC system was evaluated in treating real-field slaughterhouse wastewater while simultaneously generating bioenergy [184]. The anodic chamber of the fuel cell was inoculated with mixed bacterial species (mainly *Pseudomonas* sp. and *Bacillus* sp.), while *C. vulgaris* was used in the cathodic section. The results showed removal efficiencies of 99.65% for COD and 70% for ammonium ions from the slaughterhouse wastewater, and a maximum power output of 543.28 mW m^−2^.

### 3.10. Biopolymers

The swift pace of global industrialization, which currently poses a risk to worldwide stability, requires immediate and utmost focus on the substitution of petroleum-based commodities and polymers. Furthermore, the accumulating plastic waste in the environment, along with societies’ lack of efficient management strategies and the consequential escalation in GHG emissions, presents an urgent demand for environmentally sustainable materials [188,189].

Bioplastics, as classified by the European Union, are composed of materials from biomass or biodegradable substances [190]. Amongst the most common bioplastics within this category are several polyesters, such as polylactic acid (PLA), derived from lactic acid, poly (hydroxy alkanoate) esters (PHAs), and polyhydroxybutyrate (PHB) synthesized by a broad array of microorganisms [191]. The utilization of microalgae species for biopolymer production presents a promising alternative approach to bioplastic manufacturing. The quest for replacements to the traditional fossil-based plastic products has given birth to the notion of algae-based biopolymer production, an environmentally sound and feasible concept [191].

Recent statistics for the European bioplastic market predict a rise in bioplastic production from 2.21 million tons in 2022 to 6.29 million tons by 2027 [192]. Currently, about 100 distinct strains of eukaryotic and prokaryotic (cyanobacterial) microalgae are known to accumulate in PHB photo autotrophically, with concentrations ranging between 0.04% and 80% of their dry mass. *Chlorella* sp. and *Spirulina* (*Arthrospira*) sp. demonstrate promise in biopolymer production [193,194,195,196].

*Spirulina* sp. has been studied for PHA production by induced nitrogen deficiency. For instance, Coelho et al. [197] and Costa et al. [198] have studied PHA production from *Spirulina* sp., showing yields of 30.7% PHA (w/w dry biomass) and 12.0% PHA (*w*/*w* dry biomass), respectively. Other authors [199] have reported that the addition of acetate and CO_2_ to *S. platensis* showed an increase in PHA production. In a 2021 study conducted by Corrêa et al. [200], *A. platensis* was also used for PHA production. The study evaluated the effect of pure and crude glycerol supplementation and nutrient depletion (N and P) on PHA content in *A. platensis*. The highest PHA content was 1.1%.

In a research study conducted by Arun et al. [201], de-oiled cake obtained after algal oil extraction from *C. vulgaris* was used for PHB production in an innovative and sustainable approach. The resulting de-oiled cake showed carbon, hydrogen, nitrogen, sulfur, and oxygen compositions of 69.5, 5.2, 2.3, 0.2 and 22.8 wt%, respectively. Then, the de-oiled cake was used as the carbon source for the production of biopolymer PHB, yielding 0.41 g PHB/g of de-oiled cake. In another study, Kumari et al. [202] evaluated nature-inspired PHB production using the microalgae *C. sorokiniana*. The accumulation of microalgal PHB was driven by nutrient limitation, producing a maximum of 29.5% PHB from 0.94 mg L^−1^ of biomass. Fluorescence microscopy exposed PHB granules within the cell cytoplasm, while thermogravimetric analysis (TGA) affirmed the structure. This analysis confirmed that the biopolymer generated was a homopolymer of PHB.

### 3.11. Bioremediation

As the world’s population reaches 8 billion in 2023, the demand for fresh water, food, and energy has led to a global wastewater production rate of 400 km^3^/year [203,204]. Domestic sectors account for 70% of this, while manufacturing sectors contribute the remaining 30%. This escalating production of wastewater from both domestic and industrial sources is driving a pressing need for innovative wastewater treatment processes, spurred by concerns over water resource crises and environmental pollution [85].

Untreated wastewaters, overloaded with a diverse mix of organic and inorganic compounds, pose environmental challenges. Traditional treatment systems, while often expensive and energy-intensive, still fall short in addressing all wastewater-related issues [84]. Conventional nutrient removal methods, such as aerobic-activated sludge-based processes and chemical phosphorus removal, struggle to meet the strict discharge standards efficiently and cost-effectively [205].

Microalgae, known for their excellent biosorption capacity, have been used in municipal and industrial wastewater treatment for years due to their high biomass growth rates and resilience in various environmental conditions [105]. Their role in removing organic and emerging contaminants and recovering important nutrients such as phosphorus and nitrogen from secondary effluents, has been increasingly recognized as a sustainable strategy with industrial potential [206,207]. Given the rise in pesticide use due to growing population and nutritional needs, microalgae-enabled bioremediation offers an innovative solution for treating agricultural run-offs [208].

Many examples of wastewater treatment by *Chlorella* sp. and *Spirulina* (*Arthrospira*) sp. have been published, as these can treat many types of wastewaters, including municipal, aquaculture, swine, olive oil milling, distillery, confectionary, brine tapioca, tofu, rubber, paper mill, dye wastewaters, agricultural run-offs, and groundwaters [203,209,210,211,212]. While treating wastewater, biomass from *Chlorella* sp. and *Spirulina* (*Arthrospira*) sp. can be produced and subsequently valorized. For instance, the cells of *C. vulgaris* removed up to 93.8, 73.1, 80.5, and 85.2% of the N-NH_4_^+^, N-NO_3_^−^, P-PO_4_^3−^, and COD present in untreated urban wastewaters, while the biomass productivity achieved was similar to that obtained by cultivating *C. vulgaris* in freshwater supplemented with synthetic chemicals (0.58 g·L^−1^·day^−1^) [213].

It should also be noted that microalgae-based wastewater systems differ from the conventional wastewater treatment processes since these systems are a low-cost process with two key benefits: (1) purifying wastewater and (2) simultaneous biomass harvesting [214]. In addition, a recent study showed that it was possible to reuse the microalgae-treated wastewater for irrigation, considered the main use of freshwater, which is very economically and environmentally beneficial [215].

## 4. Challenges and Future Perspectives

*Chlorella* sp. and *Spirulina* (*Arthrospira*) sp. are sources of valuable compounds including lipids, sterols, vitamins, pigments, proteins, organic acids, polyphenols, sugars, polysaccharides, phytohormones, polyhydroxyalkonates, and more. Because of this diverse array of compounds, *Chlorella* sp. and *Spirulina* (*Arthrospira*) sp. find wide-ranging applications in human and animal nutrition, cosmetics, agriculture, pharmaceutics, medicine, tissue engineering, biofuels, photovoltaics, bioplastics, and bioremediation. Moreover, the cultivation of *Chlorella* sp. and *Spirulina* (*Arthrospira*) sp. at an industrial scale is already well established, fulfilling a pivotal role in carbon sequestration and contributing to the mitigation of climate change. Notably, their cultivation does not compete with terrestrial crops for arable land, and it can be efficiently carried out using water recirculation or even wastewaters [1,52]. As microorganisms, *Chlorella* sp. and *Spirulina* (*Arthrospira*) sp. exhibit a robust environmental adaptation ability, showcasing a relatively rapid biomass production under favorable conditions. Furthermore, their genetic manipulation offers tremendous opportunities to produce desired molecules with significant economic value. Overall, *Chlorella* sp. and *Spirulina* (*Arthrospira*) sp. present tremendous potential for sustainable and environmentally friendly applications across various industries.

Despite the considerable potential applications and research efforts, the commercial viability of using *Chlorella* sp. and *Spirulina* (*Arthrospira*) sp. biomass as a source of valuable products still encounters economic challenges due to several issues. The main concern is the low biomass productivity achieved even under optimized growth conditions [1]. Additionally, the processing of the biomass, involving extraction, isolation, and purification steps, is often complex and costly. These factors significantly increase the prices of existing large-scale produced *Chlorella* sp. and *Spirulina* (*Arthrospira*) sp., and their extracts are still higher than other similar function products. To achieve economic sustainability in the industrial use of *Chlorella* sp. and *Spirulina* (*Arthrospira*) sp., several approaches should be considered, including [1,17,51,57,216,217,218]: (1) the cultivation of organic wastes and wastewaters to reduce the demand for conventional cultivation inputs; (2) the up-cycling of side streams to recover nutrients and reduce emissions, which is the philosophy in circular bioeconomy; (3) the optimization of bioreactor designs to enhance biomass productivity and lower the operational costs; (4) the research and development of automated systems to optimize the control of the production process, as the biomass and specific metabolites strongly depend on cultivation conditions and further processing methods; (5) the application of “-omics” methodologies (genomics, transcriptomics, proteomics, metabolomics, metagenomics, and metatranscriptomics) and “high-tech” methodologies such as artificial intelligence, machine learning, computing modeling, and multivariate in the research and optimization of microalgal cultivation and processing, as well as to understand the pathways and mechanisms involved in the production of specific products; (6) the genetic engineering of strains to improve biomass and metabolite production and tailor characteristics; and (7) the biorefinery approach to extract multiple metabolites simultaneously from the biomass. By capitalizing on the potential of other high-value metabolic products and implementing these strategies, the economic feasibility of *Chlorella* sp. and *Spirulina* (*Arthrospira*) sp. use in various industries, including new ones, could be significantly enhanced.

In addition to the cost of producing and processing *Chlorella* sp. and *Spirulina* (*Arthrospira*) sp., which affect all market segments, specific problems for each market sector mentioned in the text are described below.

Nowadays, *Chlorella* sp. and *Spirulina* (*Arthrospira*) sp. have enormous applications in the human food, beverages, and nutrition sectors, and it is predicted that this scenario will continue in the upcoming years. The growth potential is mainly due to the richness of microalgae in nutrients, such as proteins, omega-3 fatty acids, vitamins, antioxidants, and minerals, holding great potential to address malnutrition and support human health and wellness. *Chlorella* sp. and *Spirulina* (*Arthrospira*) sp. also have applications in aquaculture and for aquarists and have significant potential for application in the animal feed market, serving as a feed supplement or partial substitute for common feed sources. Providing valuable basic nutrients, prebiotics, growth factors, and more, the mentioned microalgae can increase the nutritional value of the animal products (including aquatic animals), promote their physiology and health, and lead to a more sustainable animal/aquaculture production system. However, some major obstacles concerning the consumption of microalgae as functional food and feed include accessibility, availability, sustainable techniques for the extraction and purification of its metabolites, functional structure preservation during and after extraction and/or in a multidimensional food matrix, the bioavailability of the extracted metabolites in the human/animal gastrointestinal tract upon consumption, and the safety and sensory quality of the formulated food/feed products [26,219]. Furthermore, it is necessary to keep constant the nutrient content as it changes with cultivation conditions and the processing methods applied [17]. Comprehensive safety assessments are needed to ensure the absence of pollutants, toxins, and harmful bacteria in microalgae biomass, especially if cultivated in wastewaters [51,52]. A strict monitoring of the production process is key for improving the quality and safety of Chlorella and Spirulina products. In the particular case of the animals (including the aquatic ones), extensive work is necessary to understand the effects of *Chlorella* sp. and *Spirulina* (*Arthrospira*) sp. on overall health, digestion, impact on gut microbiota, regulation of oxidative stress, and overall performance [38].

*Chlorella* sp. and *Spirulina* (*Arthrospira*) sp. are gaining increasing attention in the agricultural sector due to their potential to enhance plant growth, improve soil health, and contribute to sustainable farming practices. Technological challenges and knowledge gaps limit their widespread use and incorporation into agricultural practices, including [79,80]: (1) the identification of suitable microalgal strains and combinations for enhancing plant growth and soil health through testing in various agricultural conditions; (2) advancing technology to achieve cost-effective the large-scale production, preservation, and transportation of high-quality microalgal inoculum; (3) the evaluation of different methods and timing of microalgal application in agricultural practices; (4) the identification of low-cost and efficient carriers for deploying microalgae in agriculture and biofertilizer storage; (5) the development of biofertilizers that promote plant growth, soil enrichment, and provide protection against pests and diseases; (6) studies about the effects of microalgal inoculations on soil microbial communities and their impact on plant growth and ecosystem functioning; (7) a techno-economic analysis and estimation of the environmental impacts associated with microalgal soil amendments; (8) assessment of the potential of microalgae for soil reclamation and bioremediation in saline and degraded soils; and (9) the advanced applications of microalgae for drylands to combat erosion, retain moisture, and prevent desertification.

*Chlorella* sp. and *Spirulina* (*Arthrospira*) sp. are emerging as game changers in the cosmetic and skincare industries, and the market for microalgae-based cosmetic products is expected to grow further. Their unique properties and lack of harmful side effects make them attractive alternatives to the chemically synthesized molecules currently used in beauty products [58]. The cosmetic sector is constantly in search of new ingredients to formulate innovative products [61]. Recently, there has been a great interest in cosmeceutical products, which are cosmetic formulations containing biologically active ingredients that claim medical or drug-like benefits. This trend blurs the lines between traditional cosmetics and pharmaceuticals, as these products offer a unique combination of beauty and therapeutic properties. Because of the fact that the cosmetic and pharmaceutical sectors are primarily based on the bioactive compounds from *Chlorella* sp. or *Spirulina* (*Arthrospira*) sp., the main problems that these sectors face end up being the same, and they include [57,58,59,60,61,95]: (1) further research to identify and evaluate the most promising microalgae strains for producing bioactive compounds that are biologically equivalent to existing chemicals, as well as their mechanisms of action; (2) studies on the tracing of bioactive compounds, such as vitamins, trace elements, minerals and some secondary metabolites, that are few in the literature; (3) extensive clinical trials to demonstrate the efficacy and safety of microalgae-derived products, ensuring regulatory compliance and consumer trust; (4) extensive and comprehensive studies to identify the variations in potency and functionality among the major bioactive compounds and whether they have synergistic effects on the mitigation of various diseases; (5) an evaluation on the administration methods, dosage, and effectiveness of bioactive compounds extracted from microalgae; and (6) a large-scale extraction and purification of specific compounds to target particular effects, that must be oriented to “green” and economic technologies that need to be more explored. These challenges are the same as those encountered for the healing of bioactive compounds, which find greater application in health and medicine.

In the realm of health and medicine, the potential of microalgae in chronic wound healing presents a promising avenue for future research, mainly due to the production of healing bioactive compounds—shown to have antioxidant, anticoagulant, anti-inflammatory, and antimicrobial effects. Moreover, the development of “breathing” biomaterials to address hypoxia-related complications is yet another promising prospect. Other innovative applications have also arisen, mainly the targeted delivery of drugs and oxygen by engineered *Chlorella* sp. And *Spirulina* (*Arthrospira*) sp. For enhanced chemotherapy and immunotherapy. Additionally, the engineered microalgae, with its excellent fluorescence and photoacoustic imaging properties, opens up new possibilities for the self-monitoring of the tumor therapy and tumor microenvironment. All in all, *Chlorella* sp. and *Spirulina* (*Arthrospira*) sp. have wide health applications. Thus, the exploration and utilization of these microalgae species for innovative treatments of diseases, including different types of cancers, heart diseases, and viral infections, are areas ripe for future research. However, the future of both *Chlorella* sp. and *Spirulina* (*Arthrospira*) sp. faces a significant challenge in developing safe and sustainable solutions for *in vivo* operations. To successfully bring these systems into clinical practice, collaboration among scientists, engineers, and physicians is essential [105]. Efficient energy sources or remote-control methods, possibly integrating artificial intelligence with robotics, will be crucial for the operation of microrobotic devices. Recent advancements in functional materials and device miniaturization are driving deep research in this area. As photosynthetic organisms, *Chlorella* sp. and *Spirulina* (*Arthrospira*) sp. require specific light wavelengths for growth, which may be limited within tissues [105]. Implantable technological solutions providing continuous light without causing infection or inflammation are needed for successful tissue hypoxia treatment using microalgal systems. Research should focus on enhancing the oxygen production efficiency of microalgae to further increase the local oxygen concentration at the wound site [118]. Additionally, there is a need for further exploration of the activation mode of microalgae as internal organs cannot be irradiated. Furthermore, as foreign organisms, microalgae may trigger immune responses and inflammation, which could be reduced by using coating materials [105,118]. On the other hand, the immunogenicity and viability of materials within living bodies, their capability for effective accumulation at target sites, and methods to maintain the desired local oxygen concentrations should be studied to address the current limitations and gaps in the knowledge [118]. Opportunities for integrating the multi-modal therapeutic aspects and their compatibility with photosynthetic organisms need to be explored and optimized to improve treatment efficacy [119]. Finally, proper standardization and regulation is necessary for the application of *Chlorella* sp. and *Spirulina* (*Arthrospira*) sp. in health and medicine.

In the energy sector, *Chlorella* sp. and *Spirulina* (*Arthrospira*) sp. have attracted great interest with their potential to produce a high content of lipids and carbohydrates that can be converted into a variety of biofuels. Solar energy is another innovative energy avenue for microalgae. Photosensitizers derived from *Chlorella* sp. and *Spirulina* (*Arthrospira*) sp. could potentially be used to enhance the efficiency of solar cells by absorbing light and transferring the energy to the semiconductor material in the solar cell, thereby increasing the overall efficiency of the solar energy conversion process. Another way for *Chlorella* sp. and *Spirulina* (*Arthrospira*) sp. to contribute to the production of electricity is through their applications in fuel cells. The high cost of producing biofuels and bioenergy from microalgae still poses a barrier in the efficient development of this sector [151,172,174,176]. Genetic engineering and bioengineering could potentially increase the content of energetic molecules (lipids or starch), or the production of biohydrogen, and improve the stability and photoconversion efficiency of the pigments.

*Chlorella* sp. and *Spirulina* (*Arthrospira*) sp. have shown promise for biopolymer production since they can synthesize and accumulate biopolymers, such as polyhydroxyalkanoates. These biopolymers are biodegradable and can serve as eco-friendly alternatives to petroleum-based plastics. However, the transition from laboratory to industrial-scale fabrication encounters some difficulties such as the optimization of the process conditions to obtain uniform and stable fibers [220]. Additionally, the mechanical strength and long-term stability still need to be optimized, the selection of economical and more suitable natural polymer/solvent systems require more research, and the fabrication costs need to be considered.

Microalgae have also proven to be an effective choice for municipal and industrial wastewater treatment due to their high biomass growth rates and resilience in various environmental conditions. Their role in removing organic and emerging contaminants and recovering important nutrients such as phosphorus and nitrogen from secondary effluents, has been increasingly recognized as a sustainable strategy with industrial potential and should be further exploited.

The road ahead for *Chlorella* sp. and *Spirulina* (*Arthrospira*) sp. applications is paved with opportunities and challenges. With ongoing research, multidisciplinary synergy, technological advancements, and a commitment to sustainability, these microalgae offer the potential to revolutionize several distinct segments of the market, offering effective, natural, and eco-friendly solutions for companies and consumers worldwide.

## 5. Conclusions

*Chlorella* sp. and *Spirulina* (*Arthrospira*) sp. offer significant potential for sustainable and environmentally friendly solutions across diverse industries. However, to fully harness their benefits, it is essential to overcome economic and technological challenges, as well as address specific market concerns. One primary obstacle is the high costs associated with large-scale microalgal cultivation, necessitating a focus on cost-effectiveness to achieve a broader market penetration. Additionally, each application field mentioned in this review faces its own technical constraints that require interdisciplinary collaboration, further research, and advanced technologies to be effectively overcome. By working together and advancing our understanding, we can successfully expand the large-scale production of valuable compounds from *Chlorella* sp. and *Spirulina* (*Arthrospira*) sp.

## Figures and Tables

**Table 1 bioengineering-10-00955-t001:** Nutrient composition of Chlorella and Spirulina [17,21,22,23,24,25,26,27,28].

Nutrient Composition	Chlorella	Spirulina
**Macronutrient (% dry weight)**		
Protein	42–65.5	52–72
Carbohydrate	8.1–65	9–25
Lipid/fat	1.6–40	1–8
Fiber	1.6–6	2–18
Minerals/Ash	6.3–27.3	3–13
**Essential amino acids (mg/g protein)**		
Leucine	40–95	56–84
Phenylalanine	20–96	29–48
Lysine	35–82	35–51
Valine	28–78	29–54
Isoleucine	1.0–44	1.2–41
Threonine	40–62	30–62
Histidine	10–35	6.0–28
Methionine	6.0–58	16–28
Tryptophan	1.0–24	10–20
**Other amino acids (mg/g protein)**		
Aspartic acid	38–109	54–118
Serine	13–95	23–68
Glutamic acid	76–137	70–105
Glycine	60–105	39–78
Alanine	82–159	51–108
Cysteine	2.0–35	2.0–6.0
Tyrosine	13–84	30–48
Arginine	47–74	4.0–77
Ornithine	1.2–1.3	nr
Proline	27–85	20–41
**Fatty acids (FA)**		
Saturated	25–33 ^1^	45–56 ^1^
		63–66 ^2^
Unsaturated	60–70 ^1^	41–52 ^1^
		33.8–37.1 ^2^
PUFA	36–65 ^1^	30–42 ^1^
		23.1–24.5 ^2^
ω-3		0.1–0.22
Alpha-linolenic acid (essential FA)	14–19.3 ^1^	nr
ω-6		23.1–24.5 ^2^
Linoleic acid (essential FA)	11–21 ^1^	16–17 ^1^
**Vitamins (mg/100 g)**		
B1 (Thiamine)	1.5–2.4	3.5
B2 (Riboflavin)	4.8–6.0	3.2
B3 (Niacin)	23.8	12.1
B5 (Pantothenic acid)	1.3	0.4–25
B6 (Pyridoxine)	1.0–1.7	0.78
B7 (Biotin)	191.6	64
B9 (Folic acid)	0.61–26.9	0.033
B12 (Cobalamin)	0.1–125.9	0.012–0.24
C (Ascorbic acid)	15.6–100.0	nr
E (Tocopherol)	6.0–2787.0	2.8–75
A (Retinol)	13.2	nr
K	0.033	2

^1^ % of the total fatty acids; ^2^ % of the total lipids; nr—not referenced.

**Table 3 bioengineering-10-00955-t003:** Bioactive compounds found in *Chlorella* sp. and *Spirulina* (*Arthrospira*) sp. that are most relevant for cosmetic applications, and their corresponding activities [57,58].

Bioactive Compound	Biological Activity
Carotenoids	Scavenges free radicals, fights wrinkles, delays aging, soothes eye skin. Antioxidant, anti-inflammatory. Provides blue light and UV protection. β-Carotene serves as a natural colorant in cosmetics. Lutein promotes regeneration of normal retinal blood vessels.
Vitamin C	Prevents melanin deposits, whitens the skin. Repairs the skin barrier, capillaries, and photo-aging skin, reduces erythema and telangiectasia, and lightens skin wrinkles. Stimulates collagen synthesis in the skin.
Vitamin E	Antioxidant. Repairs the skin barrier, treats some skin diseases.
Polysaccharides	Antioxidant, antibacterial. Good film-forming properties, reduces water evaporation on the skin surface and provides a moisturizing effect.
Peptides	Anti-inflammatory. Protects skin , reduces UVB and UVC-effects.
Flavonoids Phenols	Antioxidant activity. Stimulates collagen synthesis in the skin, reduces wrinkle formation.

**Table 4 bioengineering-10-00955-t004:** Examples of cosmetics marketed by major cosmetic companies.

Manufacturer	Product	Ingredient	Ref.
Estée Lauder	Perfectionist Pro	*Chlorella vulgaris* extract	[62]
Thalgo	Activ Refining Blocker	*C. vulgaris* extract	[63]
	Spiruline Boost collection (booster concentrate, antipollution gel-cream, detoxifying serum, and booster shot mask)	*Spirulina platensis* extract	[64,65,66,67]
Institut Esthederm	Intensive Spiruline collection (serum and crème)	*Spirulina maxima* extract	[68,69]
Nuxe	Merveillance LIFT collection (night cream, firming cream, lift eye cream, and firming-activating serum)	*C. vulgaris* oil	[70]
Algenist	Blue Algae Vitamin C™ Dark Spot	Blue vitamin C, phycocyanin extracted from *S. platensis* extract	[71]

**Table 5 bioengineering-10-00955-t005:** Main bioactive compounds found in *Chlorella* sp. and *Spirulina* (*Arthrospira*) sp. and their biological activities.

Bioactive Compound	Biological Activity	Ref.
Carotenoids	Antioxidant activities. Anticancer properties (oral, bladder, colon cancers; leukemia; hepatocellular carcinoma)	[30,95,96,97]
Polysaccharides	Antioxidant and antiviral activities. Anticancer properties (breast, ovary, skin, lung, colon, kidney, stomach cancers). Immunomodulation. Anti-hyperlipidemia. Neuroprotection. Anti-asthmatic effect.	[95,98,99,100]
Fatty acids	Antioxidant and antimicrobial activities. Reduces cancer risk. Heart protective properties. Treatment of arthritis. Migration and proliferation of skin cells by angiogenesis.	[30,95,99,101,102,103]
Proteins, peptides	Anticancer properties (reduces the proliferation of different lineages of neoplastic cells). Antioxidant activities.	[94,95,104]
Vitamins	Antioxidant activities.	[95]
Phenolics and flavonoids	Antimicrobial and antioxidant activities.	[25]
Found exclusively in *Spirulina* (*Arthrospira*) sp.:		
Phycocyanin	Anticancer properties (inhibits the proliferation of tumor cells, triggers cell cycle arrest, and induces apoptosis via different signaling pathways). Antioxidant, antiangiogenic, antidiabetic, and anti-inflammatory activities. Antiviral activity (inhibits the replication of several viruses such as influenza, mumps, and HIV). Immunity booster. Detoxifier.	[30,94,99,104]
Cyanovirin-N sulpholipids	Antiviral properties.	[30]
Calcium spirulan	Anticancer, antiviral activities. Immunity enhancer. Induces haematopoiesis.	[30]

**Table 6 bioengineering-10-00955-t006:** Wound healing studies based on *Chlorella* sp. and *Spirulina* (*Arthrospira*) sp.

Microalgae Species	Main Findings	Ref.
*Arthrospira platensis*	A. platensis antioxidant peptide-packed electrospun chitosan/poly (vinyl alcohol) nanofibrous mat showed no cytotoxicity in human blood leucocytes nor in the NIH-3T3 mouse embryonic fibroblast cells. It promoted fast wound healing in 3T3 cells *in vitro*, via inducing mouse embryonic fibroblast proliferation.	[107]
*Spirulina platensis*	*in vivo* assays confirmed the potential of the Spirulina- polycaprolactone (PCL) nanofibers in regenerating wounds	[108]
*S. platensis*	PCL-Alginate-Spirulina nanofiber showed no cytotoxicity towards human epithelial cells	[109]
*S. platensis*	Nanoliposomal peptides derived from *S. platensis* protein accelerated full-thickness wound healing *in vivo*, via angiogenesis and collagen production.	[110]
*S. platensis*	A Spirulina-nanophytosomal gel showed higher wound closure potential and enhanced histopathological alterations as compared to the control.	[111]
*Spirulina maxima*	Pectin derived from S. maxima showed great potential in wound healing	[112]
*Chlorella* sp.	hydrogels against diabetic wounds. In vivo outcomes confirmed *Chlorella* sp. could ameliorate hypoxia, high-glucose, excessive-reactive oxygen species, and chronic inflammation.	[113]
*Chlorella sorokiniana*	Chlorella-loaded hydrogel scaffolds, applied for 14 days on excisional wounds in mice, exhibited excellent biocompatibility in addition to significant antibacterial activity against *Escherichia coli* (99%) and *Staphylococcus aureus* (98%).	[114]
*Chlorella vulgaris*	Hydrogel material was used on wounded mice and showed pro-healing and anti-inflammatory properties	[115]
*C. vulgaris*	Chlorella extract-based hydrogel showed activity against *S. aureus* and *E. coli*.	[116]
*C. vulgaris*	Chlorella ointment containing 15% extract gives the best results in accelerating the wound-healing process and increasing the number of fibroblast cells in the soft tissue of pig ears.	[117]

**Table 7 bioengineering-10-00955-t007:** Maximum electric power (P_max_) generated by *Chlorella* sp., *Arthrospira* sp., and *Spirulina* sp. in microalgae-assisted microbial biofuel cells.

Microalgae	P_max_ (mW m^−2^)	P_max_ (mW m^−3^)	Ref.
*Arthrospira maxima*	-	100	[179]
*Chlorella* sp.	54.48	-	[181]
*Chlorella* sp.	36.4	-	[182]
*Chlorella* sp.	6.4		[182]
*Chlorella vulgaris*	13.5	-	[179]
*C. vulgaris*	18.7	-	[179]
*C. vulgaris*	-	2485	[179]
*C. vulgaris*	68	-	[179]
*C. vulgaris*	38	-	[179]
*C. vulgaris*	48.5	-	[179]
*C. vulgaris*	217.04	-	[183]
*C. vulgaris*	980	2770	[179]
*C. vulgaris*	-	3700	[179]
*C. vulgaris*	15	370	[179]
*C. vulgaris*	543.28	-	[184]
*C. vulgaris*	-	123	[182]
*C. vulgaris*	1926	-	[182]
*C. vulgaris*	24.4	-	[182]
*C. vulgaris*	248	-	[185]
*Chlorella pyrenoidosa*	30.2	120	[179]
*C. pyrenoidosa*	2.5	450	[179]
*C. pyrenoidosa*	-	99	[182]
*Spirulina platensis*	-	1.64	[179]
*S. platensis*	10	-	[179]
*S. platensis*	44.3	-	[182]
*S. platensis*	59.8	-	[186]
*S. platensis*	14.47	-	[187]

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
