# Peer review of "Emerging Applications of *Chlorella* sp. and *Spirulina* (*Arthrospira*) sp."

_bioengineering, 2023, doi:10.3390/bioengineering10080955_

Round 1

Reviewer 1 Report

The manuscript submitted by Ana P. Abreu et al. is focused on Applications of Chlorella sp. and Spirulina (Arthrospira) sp.: What´s new? Although the manuscript is well-written, I have the following comments/suggestions, which would help authors further improve their manuscript.

1.     Please revise the title; it should be more concise. It should be a sentence instead of a sentence break. I can be “Emerging applications of Chlorella sp. and Spirulina (Arthrospira) sp. It will be more attractive if you provide a graphical abstract.

2.     It is strongly recommended to improve the abstract and conclusion sections. The conclusion section should be more concise.

3.     The company name or product name can be spirulina, but when it is presented as a species name, it should be in italic.

4.     L33, L34. The information you provided is not available in reference. Please provide the correct reference.

5.     Please convert all values from (g/100g) to standard units (mg/g).

6.     L186-L190. Please provide a reference for mentioned information.

7.     L241, L242 “it has been observed that incorporating Spirulina into poultry and swine diets does not have a significant impact on animal productivity and product quality” Please provide the reference from some big journals to support your statement.

8.     L452-L469 section 3.6 Photosensitizers. The whole section is explained without reference; please provide an authentic reference to support your statement.

9.   Please carefully check and omit the grammatical and syntax errors throughout the manuscript.

The overall language of the manuscript needs to be improved to meet the standards.

Reviewer 2 Report

Please provide some figures if possible.

All full name of abbreviations should be provided when first mentioned.

The review tries to cover many topics, however this makes the whole review only cover the surface/tip of the field without much discussion. I suggest the author to select a few important topics and give more in-depth discussion. Or the author can group similar topics into one subsection and have a few big topics.

The future perspectives should be standalone section.

Reviewer 3 Report

The paper gives a good overview about the recent developments in the use of the microalgae Chlorella and Spirulina. I recommend it for publication with minor revisions. 

These refer to four remarks given below to consider:

1. On Table 2, pigment contents are given which include incorrect data with respect to zeaxanthin and astaxanthin. Zeaxanthin is not referenced for Chlorella although known to be abundant in this genus. Astaxanthin is given with 550 mg/g in Chlorella while to my knowledge it does not exist at all in this genus or if, only in traces. Due to these flaws, I would recommend that the data and references given in the tables 1 and 2 are all carefully checked and revised. Absurd data should be marked and discussed and not just cited.

2. One important aspect of the function of some of the pigments is not discussed: eye health. This is both important with respect to human consumption but also for fish larvae. With respect to the latter, it is discussed in literature that lutein improves the visibility of the prey for the larvae and thus the update efficiency resulting in higher survival rates.  

3. The aspect of the removal of micropollutants is not considered in the review. This is important because it is a precondition for that:

·       waste water can be used for microalgae production. Otherwise the biomass is contaminated 

·       microalgae are used to treat waste water to remove micropollutants which are otherwise persistent in conventional treatment plants 

·       microalgae treated waste water can be reused, which is important in arid areas

Please use reference to find out more: Reymann, T., Kerner, M., Kümmerer, K. 2020. Assessment of the biotic and abiotic elimination processes of five micropollutants during cultivation of the green microalgae Acutodesmus obliquus. Bioresource Technology Reports, DOI : 10.1016/j.biteb.2020.100512

4. Prior to publication the manuscript needs proof reading because it contains some mistakes in grammar.

The paper gives a good overview about the recent developments in the use of the microalgae Chlorella and Spirulina. I recommend it for publication with minor revisions. 

These refer to four remarks given below to consider:

1. On Table 2, pigment contents are given which include incorrect data with respect to zeaxanthin and astaxanthin. Zeaxanthin is not referenced for Chlorella although known to be abundant in this genus. Astaxanthin is given with 550 mg/g in Chlorella while to my knowledge it does not exist at all in this genus or if, only in traces. Due to these flaws, I would recommend that the data and references given in the tables 1 and 2 are all carefully checked and revised. Absurd data should be marked and discussed and not just cited.

2. One important aspect of the function of some of the pigments is not discussed: eye health. This is both important with respect to human consumption but also for fish larvae. With respect to the latter, it is discussed in literature that lutein improves the visibility of the prey for the larvae and thus the update efficiency resulting in higher survival rates.  

3. The aspect of the removal of micropollutants is not considered in the review. This is important because it is a precondition for that:

·       waste water can be used for microalgae production. Otherwise the biomass is contaminated 

·       microalgae are used to treat waste water to remove micropollutants which are otherwise persistent in conventional treatment plants 

·       microalgae treated waste water can be reused, which is important in arid areas

Please use reference to find out more: Reymann, T., Kerner, M., Kümmerer, K. 2020. Assessment of the biotic and abiotic elimination processes of five micropollutants during cultivation of the green microalgae Acutodesmus obliquus. Bioresource Technology Reports, DOI : 10.1016/j.biteb.2020.100512

4. Prior to publication the manuscript needs proof reading because it contains some mistakes in grammar.

Round 2

Reviewer 2 Report

All the comments have been addressed satisfactorily.